# Preferential selection of viral escape mutants by CD8+ T cell 'sieving' of SIV reactivation from latency

Steffen S. Docken[1], Kevin McCormick[2], M. Betina Pampena[2,3], Sadia Samer[4], Emily Lindemuth[5], Mykola Pinkevych[1], Elise G. Viox[4], Yuhuang Wu[1], Timothy E. Schlub[6], Deborah Cromer[1], Brandon F. Keele[7], Mirko Paiardini[4,8], Michael R. Betts[2,3], Katharine J. Bar[5], Miles P. Davenport[1]*

1 Infection Analytics Program, Kirby Institute, UNSW Sydney, Sydney, New South Wales, Australia,
2 Department of Microbiology, Perelman School of Medicine, University of Pennsylvania, Philadelphia, Pennsylvania, United States of America, 3 Center for AIDS Research and Institute for Immunology, Perelman School of Medicine, University of Pennsylvania, Philadelphia, Pennsylvania, United States of America,
4 Division of Microbiology and Immunology, Emory National Primate Research Center, Emory University, Atlanta, Georgia, United States of America, 5 Department of Medicine, University of Pennsylvania, Philadelphia, Pennsylvania, United States of America, 6 Sydney School of Public Health, Faculty of Medicine and Health, University of Sydney, Sydney, New South Wales, Australia, 7 AIDS and Cancer Virus Program, Frederick National Laboratory for Cancer Research, Frederick, Maryland, United States of America, 8 Department of Pathology and Laboratory Medicine, School of Medicine, Emory University, Atlanta, Georgia, United States of America

* m.davenport@unsw.edu.au

**Data Availability Statement:** Data and code used in the current study are available at the GitHub repository https://github.com/iap-sydney/CD8_sieving_TatSL8_PlosPath2023.

## Abstract

HIV rapidly rebounds after interruption of antiretroviral therapy (ART). HIV-specific CD8+ T cells may act to prevent early events in viral reactivation. However, the presence of viral immune escape mutations may limit the effect of CD8+ T cells on viral rebound. Here, we studied the impact of CD8 immune pressure on post-treatment rebound of barcoded SIV-mac293M in 14 Mamu-A*01 positive rhesus macaques that initiated ART on day 14, and subsequently underwent two analytic treatment interruptions (ATIs). Rebound following the first ATI (seven months after ART initiation) was dominated by virus that retained the wild-type sequence at the Mamu-A*01 restricted Tat-SL8 epitope. By the end of the two-month treatment interruption, the replicating virus was predominantly escaped at the Tat-SL8 epitope. Animals reinitiated ART for 3 months prior to a second treatment interruption. Time-to-rebound and viral reactivation rate were significantly slower during the second treatment interruption compared to the first. Tat-SL8 escape mutants dominated early rebound during the second treatment interruption, despite the dominance of wild-type virus in the proviral reservoir. Furthermore, the escape mutations detected early in the second treatment interruption were well predicted by those replicating at the end of the first, indicating that escape mutant virus in the second interruption originated from the latent reservoir as opposed to evolving de novo post rebound. SL8-specific CD8+ T cell levels in blood prior to the second interruption were marginally, but significantly, higher (median 0.73% vs 0.60%, p = 0.016). CD8+ T cell depletion approximately 95 days after the second treatment interruption led to the reappearance of wild-type virus. This work suggests that CD8+ T cells can actively

**Funding:** This work was supported in part by NIH grants NIAID P01AI131338 (to MPD, KJB, MRB, and MP); NHLBI, NIDDK, NIMH, NINDS, NIDA, and NIAID 1UM1AI164562-01 (to MP, MPD, and BFK); NIMH R01 MH128155, NIAID P30-AI-045008, R01 AI162646, and UM1AI164570 (to KJB). MPD is supported by an NHMRC Investigator grant (1173027) and an NHMRC Program grant (149990). This project has been funded in part with federal funds from the National Cancer Institute, National Institutes of Health, under Contract No. 75N91019D00024/HHSN2612015000003I (BFK). The content of this publication does not necessarily reflect the views or policies of the Department of Health and Human Services, nor does mention of trade names, commercial products, or organizations imply endorsement by the U.S. Government. The funders had no role in study design, data collection and analysis, decision to publish, or preparation of the manuscript.

**Competing interests:** Michael R. Betts is a paid consultant of Interius Biotherapeutics. The authors have no other competing interests to declare.

suppress the rebound of wild-type virus, leading to the dominance of escape mutant virus after treatment interruption.

## Author summary

During untreated HIV infection, CD8+ ('killer') T cells target and apply selective pressure on the HIV virus. Unfortunately, immune escape mutation may occur, allowing HIV to evade immune recognition. HIV may be more susceptible to immune pressure when there are low levels of virus, such as during initial viral rebound following treatment interruption. However, it is unknown if the immune system can target and suppress the growth of rebounding virus. In this study, we demonstrate that CD8+ T cells can inhibit the reactivation of susceptible virus following treatment interruption. However, escaped virus can circumvent CD8 pressure and reactivate if it is present in the latent viral reservoir.

## Introduction

HIV viral replication can be suppressed by ART. However, interruption of treatment leads to rebound of replicating virus in most individuals. This rebound arises from reactivation of individual 'reactivation founder' viral strains, and the frequency of reactivation from latency has been estimated to range from around once a week in humans, to several times a day in SIV-infected macaques [1–3]. This reactivation of a small number of viral clonotypes creates a population bottleneck that is potentially susceptible to immune control. Treatment with monoclonal broadly neutralizing antibodies (bNAbs) has been shown to delay viral reactivation in the absence of ART [4,5]. However, the presence of viral escape mutation limits the ability of passive antibody therapy to suppress viral replication following ATI [5–7]. Antibody escape mutants may arise from variants present in the latent proviral reservoir, and bNAbs present at the time of treatment interruption act by 'sieving' the reactivating virus and blocking susceptible variants. Alternatively, immune escape may arise after reactivation of bNAb-sensitive virus, which acquires escape mutations during post-rebound replication.

It is well established that CD8+ T cells drive immune escape during both the acute and chronic phases of HIV and SIV infection [8–13]. In the viremic SIV model, immune escape can occur very rapidly at the Mamu A*01 restricted Tat-S(T)L8 T cell epitope during the acute phase of infection [14–16]. While immune escape during fully suppressive therapy has not been demonstrated in the SIV model, studies in ART-treated people living with HIV (PLWH) have suggested that CD8+ T cell-driven viral evolution may occur [17,18]. After ATI in humans, it is also established that expansion of memory HIV-specific CD8+ T cell responses closely mirrors viral recrudescence [19–21]. Despite this association, little is known about the precise role and impact of T cell pressure on viral reactivation and growth after ATI. Previously, Okoye et al. showed that SIV-infected rhesus macaques (RMs) that underwent CD8+ T cell depletion at ATI had higher peak and set point viral loads during viral rebound than untreated controls [22], but no difference in time-to-rebound or initial growth rate of virus following treatment interruption was observed in the CD8-depleted animals. This suggests that CD8 pressure in a native, unaugmented state does not develop until after reactivation has been initiated and is unable to control early reactivation events. However, it is also important to note that the RMs in this study initiated ART early at 12 days post infection (dpi), and thus may not have developed sufficiently potent CD8+ T cell responses to suppress early events in

viral reactivation [23]. Thus, whether CD8+ T cell responses are able to suppress the reactivation or early replication of wild-type (WT) virus and act as a 'sieve' for selection of viral escape mutation has not been determined.

Here, we study the impact of CD8+ T cell pressure on immune escape following treatment interruption in a non-human primate model using barcoded SIVmac239M [3] that allows tracking of the dynamics of individual viral lineages. The barcode site is just upstream of the Tat-SL8 epitope, which has been well characterized as a locus of CD8 pressure early in SIV infection of Mamu*A01+ RMs [14–16]. Tat-SL8-specific CD8+ T cells have routinely been identified through single peptide stimulation, indicating the strong specificity of these cells [14–16], and it has been shown that addition of Tat-SL8 specific CD8+ T cells substantially inhibits SIVmac239 replication in vitro [16]. Due to the proximity of Tat-SL8 to the barcoded region of SIVmac239M, both can be sequenced simultaneously via high throughput Illumina sequencing, allowing studies of immune escape on individual viral barcode lineages. Using this strategy, we assessed the evolution of immune escape at the SL8 epitope in Mamu*A01+ RMs over the course of two periods of antiretroviral treatment followed by two treatment interruptions. We found low levels of immune escape prior to ART at 14 dpi, as well as in cell-associated SIV DNA sampled during the first period of ART suppression. After the first ATI, early rebound virus was composed of predominantly WT virus. However, we observed extensive mutation within the SL8 epitope during the first ATI. A very different pattern of viral mutation was observed during the second ATI, where escape mutant virus was dominant from the earliest timepoints. Analysis of viral barcodes and SL8 epitope mutations, along with characterisation of CD8+ T cell responses, indicated that CD8+ T cells limited the reactivation or early replication events of WT virus after the second treatment interruption. To our knowledge, this is the first evidence that CD8+ T cells are capable of inhibiting early events in the reactivation of susceptible latent virus.

## Results

### Animal experiment

14 adult male Mamu A*01+ RMs (average age 6 years) were intravenously infected with $10^4$ TCID50 SIVmac239M [3] and achieved peak viral loads of between 6.9 and 8.06 $\log_{10}$ *copies/ml*. At 14 dpi, RMs initiated daily ART (Fig 1A). Viral load was suppressed below the detection threshold of 60 copies/ml by 81 dpi in all but two animals and by 113 dpi in all animals.

ART was withdrawn on day ~220, and 13 animals had detectable rebounding virus between 10 and 25 days after ART withdrawal, while animal RNd15 did not have a detectable rebound until a single detection of virus at per-protocol ART re-initiation (61 days post ATI). Animals rebounded to mean $\log_{10}$ peak viral load of 4.54 $\log_{10}$ copies/ml (range 3.54–6.02), and then achieved a mean viral set point of 3.42 $\log_{10}$ copies/ml between days 30 and up to 60 post rebound (range 1.96–4.45; see Analysis definitions in Materials and Methods for exact time frame). The animals analysed in this study were part of a larger cohort used to determine the immunological effects of treatment with FTY720, a drug that arrests T cell migration from tissues [24, 25]. Thus, half of the animals received FTY720 for 30 days prior and throughout the first ATI (see Fig 1A). Notably, FTY720 had no significant effect on any of the viral rebound parameters used for our analyses of viral escape dynamics (similar to as reported for FTY720 treatment during early ART [26]); as a result, the FTY720-treated and control animals were combined for this study.

Next, we estimated the frequency of successful reactivation from latency after ART interruption (Fig 1G). The median reactivation rate was 3.12 reactivation events per day across all 13 reactivating animals (range 0.35 to 10.64). For these calculations, viral growth rate and

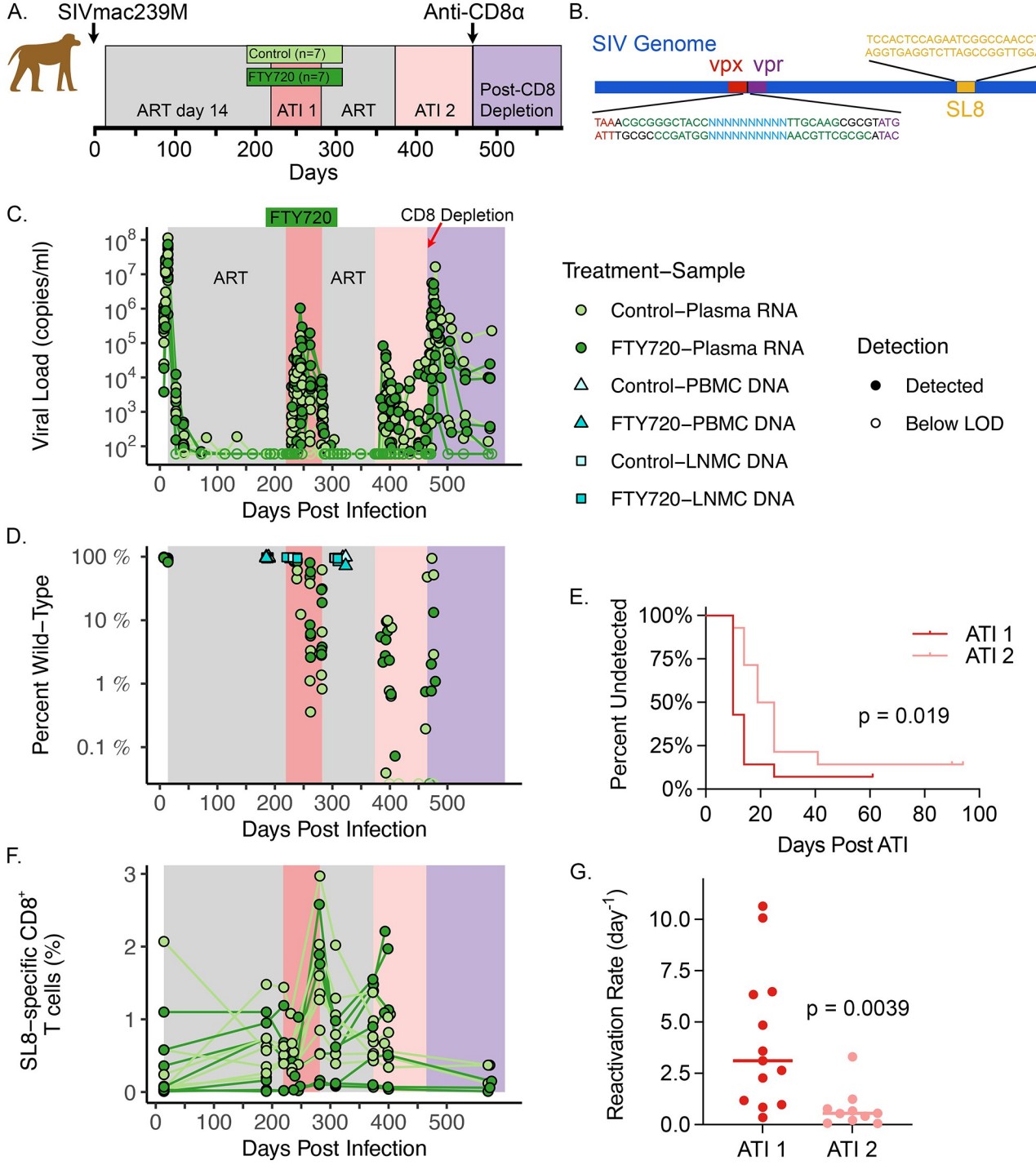

**Fig 1. Experimental Protocol.** (**A**) Schematic representation of experimental protocol. 14 animals were infected with 10⁴ TCID50 of SIVmac239M and began ART on day 14. ART treatment indicated by grey shading. 7 animals were treated during the first interruption with FTY720 and 7 animals were controls (indicated by dark green and light green bars, respectively). ART treatment was interrupted twice and 10 animals were CD8 depleted after the second treatment interruption. (**B**) As previously described [3], SIVmac239M was generated from the infectious molecular clone SIVmac239 by inserting barcodes between the vpx and vpr accessory genes, resulting in the barcodes lying 247 base pairs from the Tat-SL8 CD8 escape epitope. (**C**) Viral loads of FTY720-treated and control animals indicated by dark green and light green symbols, respectively. (**D**) Proportion of the circulating virus (dark green and light green circles for FTY720-treated and control) and LNMC and PBMC cell-associated DNA (squares and triangles, respectively; cyan and light cyan for FTY720-treated and control) that remains WT at the SL8 epitope measured by high throughput sequencing throughout the protocol. (**E**) Time-to-detection of viral rebound in first and second treatment interruptions (dark red and light red lines, respectively). P-value of log-rank test provided. (**F**) Time course of percent of total CD8+ T cells in PBMC that are specific for the Tat-SL8 epitope (dark green and light green circles

for FTY720-treated and control, respectively). (**G**) Reactivation rates based on barcode proportions (see Estimation of the reactivation rate in Materials and Methods) following first and second treatment interruptions (dark red and light red circles, respectively).

relative size of different viral barcodes within an animal were used to estimate delays between individual successful reactivations, and thereby estimate frequency of successful reactivation. This methodology (as outlined in our previous publications [2, 3] and briefly in Materials and Methods) is based on the assumptions that timing of successful reactivations within an animal are exponentially distributed and all barcodes grow at the same rate following reactivation.

A second round of ART was resumed after ~62 days of treatment interruption (at ~282 dpi), and viral load was resuppressed below detection in all animals by 21 days after ART re-initiation. Animals were maintained on the second ART treatment for 3 months until a second ATI was performed at 374 or 375 dpi. Time-to-detection of viral rebound was significantly longer at the second interruption (Fig 1E; p = 0.019). Additionally, peak viral levels during the second interruption were significantly lower than in the first interruption (median of 3.36 (n = 12) vs. 4.50 (n = 13) $\log_{10}$ copies/ml, p = 0.003), as were time weighted $\log_{10}$ viral set points (median 2.20 (n = 8) vs. 3.62 (n = 13) $\log_{10}$ copies/ml from day 30 post rebound up to day 60 (see Analysis definitions in Materials and Methods for exact time frame), p = 0.016). The frequency of successful reactivation from latency estimated using barcode analysis was around 6 times slower during the second interruption (Fig 1G), with median reactivation rate of 0.55 per day (vs. 3.12 in first ATI, p = 0.0039). A direct comparison of the number of successfully rebounding barcodes reveals the same qualitative result (p = 0.0003; see S2 Fig). We note that our sequencing only identified barcoded clonotypes that successfully expanded to the level of detection by sequencing. Therefore, it is possible many more barcodes initially reactivated from latency following ATI but subsequently failed to grow to the level of detection.

## Induction and maintenance of Tat-SL8 specific CD8+ T cells

We next assessed the longitudinal frequency and phenotype of Tat-SL8 specific CD8+ T lymphocytes in peripheral blood mononuclear cells (PBMC) via flow cytometry. We observed an induction of Tat-SL8 tetramer binding CD8+ T lymphocytes following primary infection, with a median of 0.076% (range: 0.007–2.07%; Fig 1F) of CD8+ T cells in PBMC specific for Tat-SL8 at day 14 post-infection. This was increased to a median of 0.60% by day ~190, although this was not significant (p = 0.078). The frequency of Tat-SL8 specific CD8+ T cells in PBMCs of control animals increased significantly during the first treatment interruption compared to ~190 dpi (p = 0.016), reaching peak levels of 1.35% (range: 0.52–2.97%). Note that we only consider control animals in our analysis of CD8 dynamics during the first ATI due to FTY720's effect on T cell migration in treated animals. Following initiation of the second round of ART (day 281–282), the frequency of Tat-SL8-specific T cells declined from their peak during the first interruption. When comparing the frequency of Tat-SL8-specific cells in PBMC prior to each interruption, we observed that the responses were higher prior to the second interruption (day 373–374 vs. day ~190; median 0.73% vs. 0.60% in PBMC; p = 0.016). Deeper exploration of the phenotypes of Tat-SL8-specific CD8+ T Cells in PBMC (S1 Table) revealed that although the proportion of proliferating (Ki-67+) and cytotoxic (Perforin+/granzyme B+) cells during ART were low, there was a slight decrease prior to the second ATI compared to before the first (p = 0.008 and 0.027, respectively). Thus, it appears that CD8+ T cell responses were very similar prior to the second interruption compared to the first. However, we note this is based on analysis of cells from peripheral blood samples, and it is unclear if there was a difference in the functional profile of CD8+ T cells in tissue.

## Immune escape during primary infection and treatment interruption

To investigate the development of immune escape from recognition by SL8-specific CD8+ T cells we sequenced the Tat-SL8 epitope in plasma virus and cell-associated DNA, defining all non-WT amino acid sequences as escape variants. Prior to ART initiation (14 dpi), we saw limited evidence of viral escape mutations (range 4.8–18% plasma virus was non-wild type at the SL8 epitope, circles in Fig 1D). This is consistent with previous work [27] indicating low levels of escape variants over the first 14 days of infection. Proviral cell-associated DNA in LNMC and PBMC was also analysed during the first round of ART. This analysis revealed lower levels of escape mutant virus in both LNMC and PBMC cell-associated DNA than plasma virus at time of treatment (6.0% or less in all animals at day 185–220; n = 12 for both LNMC and PBMC (squares and triangles, respectively, in Fig 1D)). This is consistent with there being very little selection for escape prior to or during the initial ART treatment, and relatively low levels of escape in the cell-associated DNA on ART.

Following the first treatment interruption, early rebounding virus (<10 days after rebound detection; n = 10) was predominantly wild-type (median 12% escape, range 3.3–55%) and the degree of escape was not significantly different from that observed during primary infection (p = 1). However, the proportion of escape mutant virus increased throughout the first ATI (Fig 2A dark red symbols), and by the time ART was re-initiated (after ~60 days of treatment

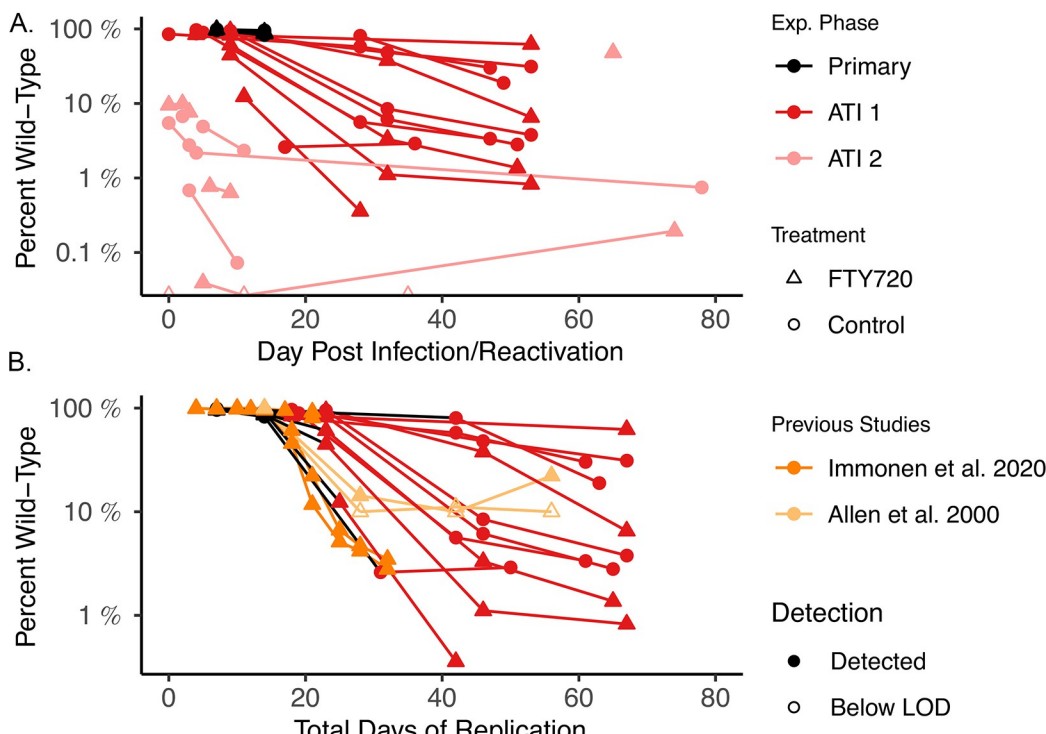

**Fig 2. Immune escape dynamics.** (**A**) Percentage of overall viral load in each animal that is wild-type at the Tat-SL8 epitope plotted vs. days post infection (black) or days after reactivation (dark and light red for interruptions 1 and 2, respectively). Control and FTY720-treated animals are indicated by circles and triangles, respectively. (**B**) Percentage of overall viral load in each animal that is wild-type at the Tat-SL8 epitope plotted vs. total days of replication (primary infection + number of days since detection in the first interruption). Time points from primary infection and 1st ATI again plotted in black and dark red, respectively. Data from Immonen et al. 2020 [27] and Allen et al. 2000 [14] are also plotted in dark and light orange, respectively (background subtraction not applied to data from previous studies and open symbols indicate limit of detection at time points when wild type was not detected in these studies).

interruption, 36–53 days post reactivation) we observed that an average 85% of plasma virus was escaped within the Tat-SL8 region (range 38–99%; n = 11). We also considered immune escape as a function of the total days of (untreated) viral replication (Fig 2B). Interestingly, despite the >200 days of ART treatment, the accumulation of escape mutant virus followed a similar trajectory to that seen in previous primary infection studies [14, 27]. Furthermore, in both this and previous studies, decline in wild-type prevalence slowed after dropping to near 10% of the viral load and then appeared to persist in some animals at approximately 1% of the plasma virus. It is unclear why wild-type virus persists at these low levels despite the clear competitive advantage of escaped variants.

Despite the high levels of viral escape at the end of the first treatment interruption, proviral cell-associated DNA in LNMC and PBMC remained predominantly wild-type during the second round of ART (Fig 1D; days 303–323), suggesting only a small proportion of the reservoir was reseeded by the dominant virus circulating at the time of ART re-initiation. On average, LNMC and PBMC proviral DNA was 94% (n = 8) and 91% (n = 4) wild type during the second round of ART (ranges: 85–97% and 72–99%, respectively).

## Reactivation of escape mutant virus early after second interruption

To investigate if prior treatment interruptions affect reactivation kinetics following subsequent ATIs, animals were recommenced on ART after the first ATI for a further 93 days, followed by a second interruption. Following the second ATI, high levels of immune escape were observed from the earliest sequencing timepoint (Figs 1D and 2A). Among animals sequenced within 6 days after rebound (n = 10) a median of 96% of virus was already escaped (range: 90% - 100%). This stands in stark contrast to the first interruption, where a median of 12% escape was observed in samples collected 10 days or less post rebound (p = 0.016).

The escape variants observed early during the second treatment interruption may have arisen from two possible sources. Firstly, wild-type virus may have reactivated from the latent reservoir, but then rapidly acquired de novo mutations and been selected due to immune pressure, leading to dominance of escape mutation in early samples. Alternatively, the escape variants generated during the first ATI may have directly reactivated from the reservoir and dominated early viral replication in the second ATI. To evaluate these hypotheses, we examined how well specific escape variants detected early in the second ATI were predicted by the corresponding recently replicating virus in the same animal (an 'escape reactivation' hypothesis). Alternatively, we considered whether WT virus may have reactivated and subsequently developed escape mutations de novo (a 'novel mutation' hypothesis). As illustrated in Fig 3, we found that variants detected early in the second ATI in each animal were best predicted by the variants replicating at the end of first ATI in the same animal (see S3 Text for details). Although the diversity of Tat-SL8 variants was broad during the first ATI, specific variants reached dominance at ART re-initiation in each animal. These dominant variants were then more likely to be detected in the rebounding viral load after the second ATI (indicated by red arrows). This strongly supports the hypothesis where escape variants in the second interruption arose from the escape variants that dominated in the first interruption.

## Inhibition of wild-type Tat-SL8 reactivation from latency

The rapid appearance of escape variants during the second ATI could have occurred due to either higher reactivation rates of escaped virus compared to WT virus during the second ATI, or due to incomplete viral suppression during the shorter duration (3 months) of the second treatment phase. However, the available evidence strongly suggests a slower rate of successful reactivation in the second interruption than the first. Reactivation during the second

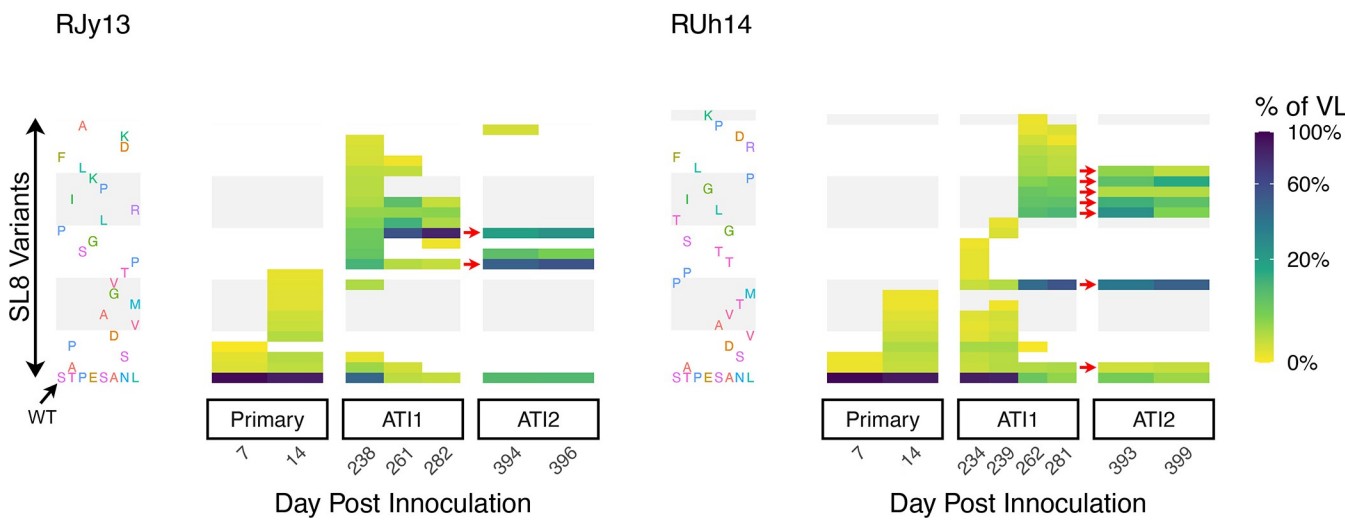

**Fig 3. Dynamics of the Tat-SL8 epitope variants.** The prevalence over time within animals RJy13 (**A**) and RUh14 (**B**) of wild-type and Tat-SL8 single amino acid variants resulting from a single non-synonymous nucleotide mutation. Each row corresponds to a different variant (wild type in the bottom row), and each column is a different sequencing time point. Percent of viral load composed of each variant indicated by colour bar. Red arrows indicate variants that were detected late in the first interruption and subsequently rebounded in the second treatment interruption.

interruption was delayed compared to the first interruption (Fig 1E; p = 0.019) and the frequency of reactivation was also around 6 times lower in the second interruption than in the first interruption (Fig 1G; median 0.55 per day vs 3.12, p = 0.0039). This argues against the possibility of a high reactivation rate of escaped virus or incomplete viral suppression. An alternative explanation, therefore, for the dominance of escape variants early in the second interruption, is immune 'sieving' of reactivating virus by Tat-SL8-specific T cells, resulting in suppression of wild-type virus rebound and thus leading to escape variants constituting the majority of reactivations observed.

To test this explanation, we performed a CD8 depletion in 10 animals following the second ATI (range: 90 to 97 days after the second ART interruption; approximately 465 days after initial infection) and assessed whether, once CD8⁺ T cell pressure was removed, WT virus was permitted to rebound. For this depletion, the anti-CD8α antibody MT807R1 was used, which also depletes Natural Killer Cells. Unsurprisingly, CD8 depletion had marked effects on plasma viral loads, including rapid increases in plasma viral levels in all animals (by mean of 1.79 $\log_{10}$ copies/ml over first 4 days post depletion; Fig 4A), and detection of viral rebound in two animals that had previously demonstrated spontaneous control of viremia.

To analyse whether CD8 depletion affected the ability of WT virus to rebound, we compared viral lineages (i.e., barcodes) detected to have rebounded prior to CD8 depletion with those barcodes that were only seen after depletion (see Analysis definitions in Materials and Methods). We found strong evidence that CD8⁺ T cell depletion radically altered the profile of rebounding virus. Firstly, there was a major difference in when the rebounding lineages were last observed in the replicating plasma virus pool (Fig 4B). Prior to CD8 depletion, the majority (64/104) of the rebounding barcodes were also present in plasma virus late in the first interruption (i.e., the most recent period of viral replication). In contrast, after CD8 depletion very few of the rebounding barcodes had been observed late in the first interruption (16/90), and indeed the majority of barcodes emerging after CD8 depletion (64/90) had been seen only in primary infection (day 7 or 14) or had not been sampled previously.

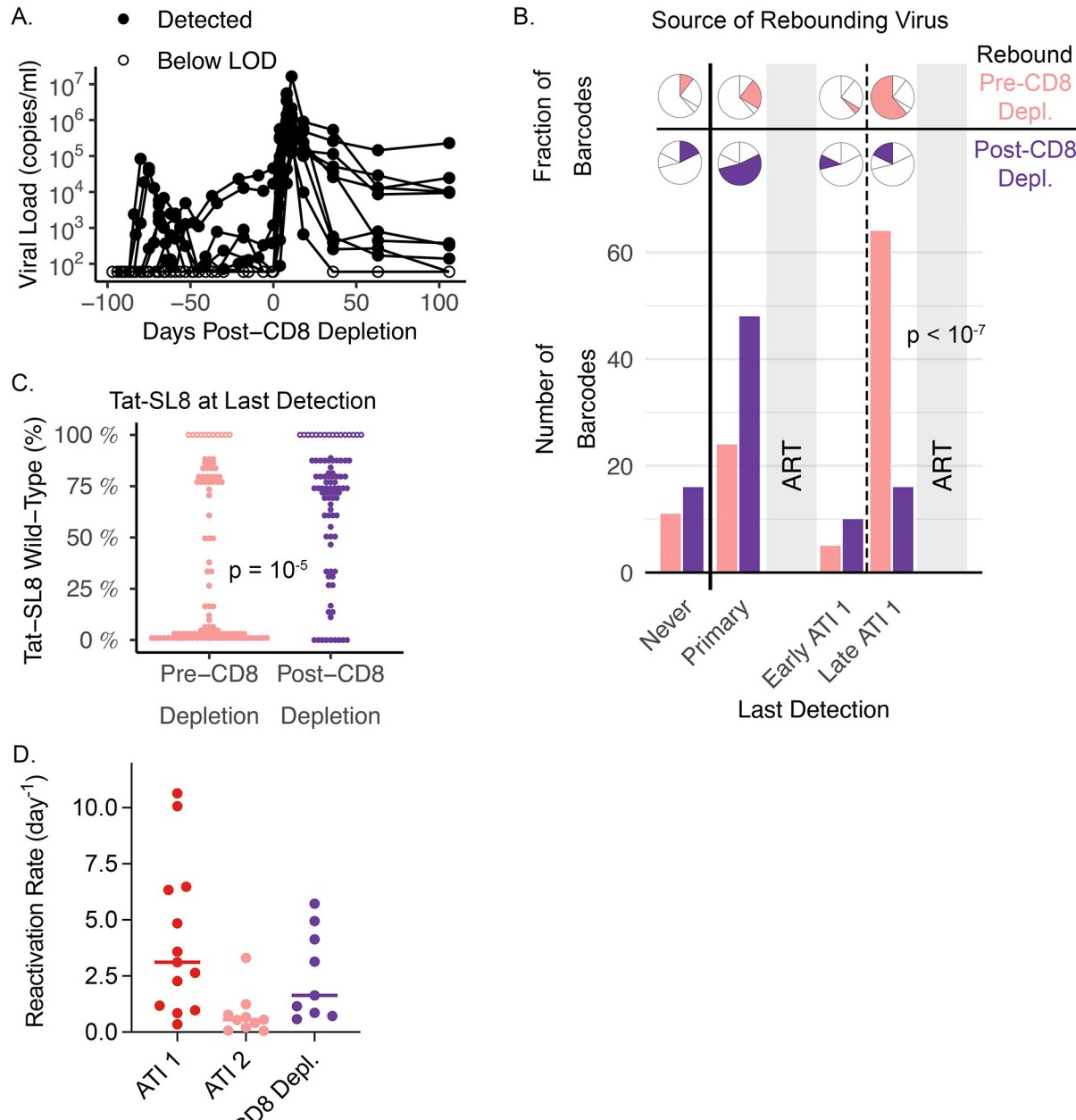

**Fig 4. Wild-type Tat-SL8 reactivates after CD8 depletion.** (**A**) Plasma viral loads pre- and post-CD8 depletion for 10 animals that were CD8 depleted during the second treatment interruption. (**B**) Breakdown of the origins of barcodes (i.e., last detection within plasma RNA) detected either before CD8 depletion or emerging after CD8 depletion. Fractional breakdown and total number for each prior time point indicated by pie charts and bar plot respectively (Pearson's chi-squared p-value provided). (**C**) Percent wild-type Tat-SL8 epitope at last detection prior to the second ATI. For barcodes never previously detected within the same animal (open circles), last detection was in the infecting inoculum, and therefore 100% wild-type Tat-SL8. P-value for association between percent wild-type at last detection and timing of detection in the second ATI is provided (Mann-Whitney test). (**D**) Reactivation rate estimates based on barcode ratios (see Estimation of the reactivation rate in Materials and Methods) for the first and second ATI, and post-CD8 depletion. For (**B-D**), barcodes seen in the second ATI that reactivated prior to and post-CD8 depletion are indicated by light red and purple, respectively.

The other major difference between the barcodes rebounding before or after CD8 depletion was whether those barcodes had previously been observed to be associated with escape mutations (in the plasma virus). Before CD8 depletion, most of rebounding barcodes had been

observed to be escaped at last detection (65/104 barcodes were >50% escaped at the Tat-SL8 epitope). However, of barcodes emerging after CD8 depletion, only a minority were associated with majority escape variants at last detection (only 21/90 were >50% escaped, Fig 4C). In other words, sequencing of plasma virus after CD8$^+$ T cell depletion revealed the appearance of novel SIV barcodes that were mainly WT at the Tat-SL8 epitope when previously replicating. This strongly suggests that rebound of these lineages had been suppressed in the presence of CD8$^+$ T cell pressure, and only released after CD8 depletion.

The evidence above indicates that the detection of WT virus was greatly reduced among rebounding virus during the second ATI, likely as a result of Tat-SL8-specific CD8$^+$ T cell pressure blocking early reactivation or replication events of WT virus. This stands in contrast to the first treatment interruption, where early viral rebound was dominated by WT virus, suggesting WT virus was able to replicate to detectable levels despite the presence of Tat-SL8-specific CD8$^+$ T cells. It is important to differentiate between CD8$^+$ T cell effects on early viral reactivation events versus later viral replication and set point. For example, although WT virus dominated the rebound during the first ATI, the set point viral load was only a mean of 3.42 log$_{10}$ copies/ml. This is substantially below the expected set point during primary infection of greater than 6 log$_{10}$ copies/ml [22, 28], suggesting substantial immune control. Okoye et al. have shown that rebound set points after 12 months of ART are around 2 logs lower than in primary infection, and that this is reversed by CD8 depletion. This suggests a major role of CD8s in controlling post-rebound set point viral levels [22]. This is also supported by the eventual dominance of escape mutant virus during the first ATI, suggesting selection pressure from Tat-SL8-specific CD8$^+$ T cells. Thus, although WT virus was able to successfully reactivate early in the first ATI, it also experienced substantial CD8$^+$ T cell pressure. This is consistent with Okoye et al.'s findings that CD8$^+$ T cell depletion during a first treatment interruption did not alter the early viral rebound kinetics but did alter the later set point viral levels [22]. In the second treatment interruption, however, Tat-SL8-specific CD8s controlled WT viral rebound to below the level of detection. The later set point level (of the predominantly EM viral population) was also lower than in the first treatment interruption (median of 2.20 log$_{10}$ copies/ml), suggesting evidence of alternative control mechanisms. The strong suppression of WT virus during the second ATI suggests there was a quantitative or qualitative change in CD8$^+$ T cell function at the start of the second treatment interruption compared to the start of the first. Analysis of SL8-specific CD8$^+$ T cell concentration and phenotype (reported above) revealed minor differences, which do not seem sufficient to explain this apparent sieving of WT viral rebound in the second ATI. However, it seems likely that the response induced by 14 days of exposure to viral replication (before initial ART treatment) induced a different Tat-SL8-specific CD8$^+$ T cell response to that seen after an additional 60 days of treatment interruption.

The suppressive effect of CD8$^+$ T cells during the second treatment interruption is supported by not only the predominance of escape mutant virus, but also by the reduction in overall reactivation rate (from a median of 3.12/day to 0.55/day; p = 0.003), consistent with a >80% reduction in rebound frequency. The rebound frequency rose again after CD8$^+$ T cell depletion and was around 3-fold faster than the reactivation rate observed earlier in the second interruption (median 1.64 vs. 0.55/day; p = 0.059). Importantly, there was no significant difference between rebound rates after the first treatment interruption and after CD8$^+$ T cell depletion (median 3.12 vs. 1.64; p > 0.99), suggesting that in both cases this reflected SIV rebound kinetics in the absence of CD8$^+$ T cell sieving. Direct comparison of the number of rebounding barcodes following ATI 1, ATI 2, and after CD8 depletion reveals similar qualitative results as our estimation of reactivation rate (see S2 Fig)

## Discussion

It has long been known that CD8+ T cells play an important role in the immune response during primary infection of PLWH and in SIV-infected RMs [29, 30]. However, the role of CD8+ T cells during latency and viral reactivation from latency following treatment interruption are unclear. Two common hypotheses are that 1) CD8+ T cells selectively restrict, or "sieve" the latent reservoir, blocking early events in the reactivation of sensitive (non-escaped) variants; or 2) CD8+ T cell responses arise after initial virus reactivation has occurred, and the resultant selection pressure selects for de novo mutants during virus replication. In this study, we sought to infer the role of CD8+ T cells during reactivation by monitoring the evolution of the Mamu-A*01-restricted Tat-SL8 epitope over the course of successive treatment interruptions. Based on the relatively low frequency of successful reactivation events following the second treatment interruption, and the fact that reactivation was dominated by escape variants seen prior to treatment, we infer that CD8+ T cells acted as a "sieve" of the latent reservoir to suppress early events in the reactivation of WT virus after the second treatment interruption. We note that the cell-associated viral DNA sampled prior to the second ATI was predominantly WT and that, despite this, escape variants dominated the observed reactivation. Given sequencing only identified clonotypes that had successfully grown to the level of detection by sequencing, other latently infected cells may have reactivated during the ATIs but either died out or failed to grow to the level of detection. It is not possible to differentiate whether CD8+ T cell "sieving" of clonotypes occurs via early killing of the first reactivating cell, or control of subsequent viral replication such that a clonotype did not reach the level of detection.

The inferred "sieving" action of CD8+ T cells is supported by the fact that subsequent CD8 depletion led to greater reactivation of the WT epitope, confirming the presence and competence of WT virus in the reactivating reservoir when not exposed to CD8 activity. It remains unclear why CD8+ T cells appeared to permit the rebound of WT virus during the first ATI but appeared to suppress it so successfully during the second ATI. One possibility is that primary infection allowed only 14 days of viral replication before ART, and that this did not provide sufficient stimulation (and maturation) of responding CD8s. By contrast, the 60 days of viral replication during the first ATI may have boosted the quality and/or quantity of responding SL8-specific T cells. However, although the number of SL8-specific CD8+ T cells in peripheral blood was mildly elevated prior to the second treatment interruption, this seems a surprisingly small change to have mediated suppression of WT rebound. Another possibility is that there was higher T cell activation at the time of the second interruption (for example because of the shorter (3 month) treatment). This again was not obvious from the T cell phenotype data, but comparison of phenotypes prior to ATIs was limited to peripheral blood due to sample availability. Hence, it is not clear if these data were fully representative of phenotypes present in the tissues. Further work is clearly required to delineate the numerical, phenotypic, and / or anatomical requirements that allow the SL8-specific T Cells to suppress early events in susceptible virus reactivation from latency.

Interestingly, the high frequency of immune escape from CD8 pressure during the second ATI did not lead to reduced virologic control compared to the first ATI. In fact, both peak and set point viral load were significantly lower at the second ATI than the first, potentially due to alternative means of immune pressure or the fitness costs of escape mutation [31]. Importantly, this suggests that immune escape during an ATI does not necessarily imply reduced immunologic control of viral replication during subsequent treatment interruptions.

It was previously suggested that CD8 pressure may drive viral evolution and escape during ART [17]. Our data strongly suggests that this is not the case given cell-associated proviral DNA remained predominantly wild-type during both rounds of ART and at both treatment

interruptions the viral quasispecies early after reactivation was dominated by virus previously observed in plasma.

A number of HIV studies have found that during ART, the composition of cell-associated DNA is biased towards virus that was circulating in the years prior to ART initiation [32–35]. We find that virus reactivating following treatment interruption is even more biased towards virus circulating at the time of ART (i.e., sampled within a matter of days prior to ART), as evidenced by the close agreement between barcodes and Tat-SL8 variants replicating at initiation of the second round of ART and those observed reactivating after the second treatment interruption. This was not because of a dominance of these variants in the cell-associated SIV DNA (observed during ART). Examination of the proviral DNA sequences before and after treatment interruption suggests that there was only slight reseeding of proviral DNA by replicating virus over the course of the first ATI. The vast majority of the proviral DNA observed during the second round of ART was established in primary infection (based on the high level of wild-type proviral DNA). Despite this, escape mutant virus observed during the first treatment interruption must have contributed to the rebound competent reservoir, given the strong relationship between virus circulating before and after the second round of ART (despite the fact that these mutants constitute only a minor fraction of the proviral DNA). The minimal reseeding of proviral DNA during treatment interruption is consistent with previous HIV/SIV studies [36–39]. Salantes et al. found significant differences in the composition of latent HIV-1 clones capable of producing virus on ART (i.e., results of the quantitative viral outgrowth assay on samples collected during ART) and those that were actually observed after treatment interruption (i.e., detected in replicating virus following ATI) [36], suggesting that even among replication competent proviral DNA, only a small subset is capable of initiating ongoing infection. Based on the detected rebounding of "older" virus following CD8 depletion in this study, we conjecture that susceptibility to immune pressure plays a substantial role in defining the component of proviral DNA that is most likely to initiate successful viral rebound.

One limitation of this study is that escape from CD8⁺ T cell recognition was not formally confirmed for all of the SL8 variants observed in this study. However, only 5 non-wild-type variants were ever detected at greater than 20% of the viral load in any animal (listed in S2 Table), and all of these mutants had previously been documented as escape variants and detected within Mamu-A*01⁺ RMs [14, 15, 27, 40]. We note that as stated above, CD8 depletion was performed using the anti-CD8α antibody MT807R1, which also depletes Natural Killer Cells. However, the observations made here are highly epitope-specific, and it is not apparent that Natural Killer Cells can selectively target WT vs. escape mutant virus at the SL8 epitope. It should also be noted that in the comparison of viral lineages reactivating pre- vs. post-CD8 depletion, for some animals that reactivated pre-CD8 depletion, the most recent sequencing was performed over 60 days prior to CD8 depletion. Thus, some barcodes may have reactivated in the interim, causing an overestimation of the reactivation rate. However, 6 animals either had undetectable plasma virus for at least 30 days prior to CD8 depletion or had sequencing performed less than 20 days pre-CD8 depletion, giving greater confidence in the estimates presented here.

A final limitation of this study is it only examines the Tat-SL8 epitope, whereas there are many other CD8 T cell epitopes that have been documented in SIVmac239. Both antibodies and CD8⁺ T cells specific to other epitopes likely play a role in reducing viral replication during ATI. However, it is not expected that these mechanisms, which target epitopes distinct from Tat-SL8, could disparately affect the different SL8 mutants that are the focus of our analysis. Thus, it is unlikely these other immune mechanisms were playing a role in the observed suppression of reactivation or early viral replication events of WT virus at the second interruption.

This study demonstrates that CD8 pressure can inhibit the early events in the reactivation of latent SIV/HIV, as evidenced by the slower reactivation rate and suppression of observed

WT reactivation at the second treatment interruption. In the SIV/Mamu-A*01+ NHP model used here, it seems likely that exposure to virus replication during the first ATI may have increased CD8+ T cell function (while only modestly increasing SL8-specific cell numbers) leading to more effective control of WT virus during the second ATI. Notably, the exposure to viral replication during the first ATI may have acted as a double-edged sword, with replication providing antigen that boosted CD8 responses, but also facilitated viral escape from the same CD8 responses. These escape variants then seeded the reservoir and escaped CD8+ T cell recognition at the subsequent ATI. However, this work suggests that strategies to boost SIV-specific CD8+ T cell responses during ART (without stimulating viral escape) may permit the control of (at least wild-type) virus after treatment interruption.

## Materials and methods

### Ethics statement

All procedures in this study were conducted following guidelines set forth by the Animal Welfare Act and by the NIH's Guide for the Care and Use of Laboratory Animals and performed in accordance with institutional regulations reviewed and approved by Emory University's Institutional Animal Care and Usage Committee (IACUC; Permit number: PROTO201700655) at Emory National Primate Research Center (YNPRC). Animal care facilities are accredited by the U.S. Department of Agriculture (USDA) and the Association for Assessment and Accreditation of Laboratory Animal Care (AAALAC) International. Appropriate procedures were performed to ensure that potential distress, pain, discomfort and/or injuries were limited to unavoidable level while conducting the research plan. Sedatives and analgesics were used when determined by veterinary medical staff for blood and tissue collections. Euthanasia of RMs was performed at the end of the study by veterinary medical staff, in accordance with IACUC endpoint guidelines.

### SIV infection and antiretroviral therapy

This study was conducted on a cohort of 14 male, Mamu-A*01+ RMs between 46–129 months of age at infection. RMs were infected intravenously with 10,000 TCID50 of SIVmac239M, a genetically tagged virus with a 34-base genetic barcode inserted between the vpx and vpr accessory genes of the infectious molecular clone SIVmac239 (schematically illustrated in Fig 1B) [3]. All animals initiated a potent three drug ART regimen at 14 days post infection that included tenofovir (TDF; 5.1 mg/Kg per day), emtricitabine (FTC; 40 mg/Kg per day) and dolutegravir (DTG; 2.5 mg/Kg per day) formulated in a single daily injection (1ml/Kg per day; s.c.). ART was maintained until ~220 dpi, at which point the animals were stratified based on peak viral load, age, weight, and time to ART suppression of viremia and assigned to intervention/control groups. ART was interrupted for two months from approximately day 220 to day 282 post infection. FTY720 was orally administered to half the animals (n = 7) at a dose of 500µg/Kg per day beginning 1 month prior to ART interruption for a total of four months (from ~190 to ~310 dpi). Four animals were necropsied between 392 and 403 dpi. All animals underwent a second cycle of ART lasting three months (from ~282 to ~375 dpi). Ten animals underwent CD8+ T cell depletion via a subcutaneous injection of 50mg/kg of MT807R1 antibody between 465 dpi and 471 dpi and were necropsied on day 571 or later.

### In vivo sample collection

Peripheral blood (PB) and lymph node (LN) biopsies were collected longitudinally. Blood collections were performed frequently during ATIs. Plasma was separated by density gradient

centrifugation and used for quantification of viral RNA and flash frozen and used for viral sequencing. Peripheral blood mononuclear cells (PBMCs) were isolated from the remaining blood sample using a Ficoll-Paque Premium density gradient (GE Healthcare) and cryopreserved. The LN site over the axillary or inguinal region was clipped-off followed by an incision. The LN was then exposed upon dissection and excised over clamps. Each LN was segmented; one part was flash frozen in dry ice for RNA-Seq or for studying the pro-viral compartment (Cell-associated DNA/RNA) of SIV, and the other part was mechanically disrupted through a 70 μm cell strainer and washed with R-10 media composed of RPMI 1640 (Corning) supplemented with 10% heat-inactivated fetal bovine serum (FBS), 100 IU/ml penicillin, 100 μg/ml streptomycin, and 200 mM L-glutamine (GeminiBio) to obtain mononuclear cells that were cryopreserved for flow cytometry analysis.

## Plasma viral load quantification

Plasma SIVmac239M loads were quantified regularly throughout the study at the Virology Core of the Emory Center for AIDS Research using a standard quantitative PCR (qPCR) assay (with a limit of detection of 60 copies/ml) as described previously [41].

## Barcode and Tat-SL8 sequencing

RNA was extracted from plasma samples using QIAGEN EZ1 Virus Mini Kit v2.0 and cDNA was synthesized using Superscript III reverse transcriptase and a reverse primer (SL8R1: 5'-AGC TGA GAG AGG ATT TCC TCCC-3'). DNA was extracted from mononuclear cells using QIAamp DNA Blood Mini Kit. qRT-PCR was used to quantify the number of templates using SYBR Green and primers (VpxF1: 5'-CTA GGG GAA GGA CAT GGG GCA GG-3' and VprR1: 5'-CCA GAA CCT CCA CTA CCC ATT CATC-3'). DNA was amplified and MiSeq adaptors were added directly to the amplicons using forward primer VpxF1.Illumina: 5'-TCG TCG GCA GCG TCA GAT GTG TAT AAG AGA CAG CTA GGG GAA GGA CAT GGG GCA GG-3' and reverse primer SL8R1.Illumina: 5'-GTC TCG TGG GCT CGG AGA TGT GTA TAA GAG ACA GAG CTG AGA GAG GAT TTC CTC CC-3'. PCR conditions included an initial denaturation of 2 minutes at 94˚C, followed by 40 cycles of denaturation (94˚C, 15 sec), annealing (60˚C, 1 min 30 sec), and extension (68˚C, 1 min), followed by a final extension step of 5 minutes at 68˚C. Reactions were pooled and gel extracted using QIAquick Gel Extraction Kit to remove primer dimers and sequenced with Illumina Next Generation Sequencing approach.

Sequencing errors are relatively common in Illumina Next Generation Sequencing, and therefore to be confident in the results, output reads were discarded if the barcode did not match a barcode from the previous stock characterization, the Tat-SL8 epitope was not detected, or an indel was detected in the Tat-SL8 epitope. Number of templates and sequence amplification varied between sampling time points, and sequencing runs with less than 1000 templates (200) or 200 reads (100) for RNA (DNA) after the removal of erroneous reads were discarded from further analysis. Limit of detection was set for each sequencing run based on number of templates and output reads (see S1 Text for details). Due to differences in limit of detection for various barcodes and Tat-SL8 variants at different time points, for quantification measurements, background subtraction was performed on all barcodes, Tat-SL8 variants, and barcode-Tat-SL8 combinations detected unless stated otherwise.

## T cell quantification and phenotype characterization

**MHC Class I Tat-TL8 Tetramer.** The Mamu-A*01 Tat28-35 APC conjugated tetramer was constructed using an $SIV_{MAC251}$-derived peptide (TTPESANL) and was supplied by the

NIH Tetramer core facility at Emory University, Atlanta, GA. Even though the corresponding $SIV_{MAC239}$ sequence was STPESANL, this tetramer detected strong responses in $SIV_{MAC239}$-infected macaques, as it has been shown before where $Tat_{28-35}$ STPESANL tetramer staining yielded identical results [14].

**Fluorescence cytometry staining.** The following antibodies were obtained from BD Biosciences: CD4 BUV661 (clone SK3), CD95 BUV737 (clone DX2) and CD3 BUV805 (clone SP34-2); from Biolegend: CD8 BV650 (clone RPA-T8), CD14 BV510 (clone M5E2), CD16 BV510 (clone 3G8) and CD20 BV510 (clone 2H7); from Beckman Coulter: CD28 ECD (clone CD28.2). Briefly, cryopreserved PBMC were thawed and stained with MHC-Class I tetramer for 10 minutes. Then, cells were stained for viability exclusion using Live/Dead Fixable Aqua (Invitrogen) for 10 minutes, followed by a 20-minute incubation with a panel of directly conjugated monoclonal antibodies diluted in equal parts of fluorescence-activated cell sorting (FACS) buffer (PBS containing 0.1% sodium azide and 1% bovine serum albumin) and Brilliant stain buffer (BD Biosciences, San Jose, CA). Cells were washed in FACS buffer and then fixed/permeabilized using the FoxP3 Transcription Factor Buffer Kit (eBioscience, San Diego, CA). Intracellular staining was performed by adding the antibody cocktail prepared in 1X perm-wash buffer for 1 hour. Incubations were all done at room temperature. Stained cells were washed and fixed in PBS containing 1% paraformaldehyde (Sigma- Aldrich, St. Luis, MO). Samples were acquired using a FACS Symphony A5 cytometer and analysis was performed by Flowjo software 10.8.1 (Tree Star Inc.).

## Statistical and mathematical analysis

Unless otherwise stated in the text or following Methods, all statistical tests for comparisons of measurements at two time points were performed using the paired Wilcoxon signed rank test.

**Estimation of the reactivation rate.** In order to estimate the reactivation rate in an animal ($R$), we use the same method as in our previous work [2, 3]. Briefly, we estimate the reactivation rate using the average log-ratio of barcodes, and assuming that all barcodes have the same growth rate ($g$) during initial exponential growth of the virus and that the growth rate of a single barcode is the same as the growth rate of total viral load. We estimate the delay between reactivations of two barcodes using the growth rate and the log-ratio of the barcodes ($\ln S_2 - \ln S_1$) as $\Delta = \frac{\ln S_2 - \ln S_1}{g}$. Therefore, the estimated reactivation rate based on $n$ rebounding barcodes is the inverse of the average of $n-1$ delays between reactivations:

$$R = \frac{g(n-1)}{\sum_{i=1}^{n-1}(\ln S_{i+1} - \ln S_i)}, \tag{1}$$

where $S_i$ is the number of reads for barcode $i$, for $i = 1, 2, \ldots, n$. To estimate the viral growth rate in an animal ($g$), we first calculated the growth rate of the virus between each pair of neighboring measurements (timepoints $t_1$ and $t_{l+1}$) following rebound of virus in said animal. This included the interval from the last timepoint at which the viral load was below the detection threshold in the animal (for which viral load is approximated as detection threshold value) and the first measurement at which the viral load is above the detection threshold in. Therefore, we have:

$$g_l = \frac{\ln V(t_{l+1}) - \ln V(t_l)}{t_{l+1} - t_l}, \qquad l = 1, 2, \ldots$$

where $V(t_l)$ is the viral load at the time $t_l$. The growth rate $g$ for the animal is assumed to be the maximal growth rate between any two neighboring measurements:

$$g = \max_l g_l.$$

The procedure of estimation of reactivation rate was implemented in Wolfram Mathematica, Wolfram Research Inc, Champaign, IL, USA. The paired Wilcoxon signed rank test (GraphPad Prism 9.4.1) was used to compare reactivation rate following the first and second ATI.

For estimation of reactivation rate following CD8 depletion, the same methodology was implemented, except only including barcodes deemed to have reactivated after CD8 depletion (see Analysis definitions below). Comparison of reactivation rates across the three time points; first ATI, second ATI, and CD8 depletion, was done using the Kruskal-Wallis test with Dunn's multiple comparisons (GraphPad Prism 9.4.1).

**Estimation of time-to-detection.** We used GraphPad Prism 9.4.1 to obtain the Kaplan-Meier curves for the time to detected reactivation and compared the time to detected reactivation following the two ATIs using the log-rank test.

The time to detected reactivation was considered to be the earliest of 1) the first of at least two viral load quantifications above limit of detection or 2) viral load quantification of above 1000 copies/ml on a single day. Day post ATI was calculated by defining the first day without ART as day 0 post ATI. In the case that ART was resumed, CD8s were depleted, or animal was necropsied before reactivation was detected, animals were right censored at the day of said event.

**Predictors of plasma virus variant composition post treatment interruption.** To assess the factors contributing to the detection of a variant in plasma virus after treatment interruption, we consider different potential mechanisms for generating escape mutations in the rebounding virus; One possibility is that the initial rebounding virus is actually wild type, but that mutation and selection during early replication led to the dominance of EM virus by the time we sample plasma. In this case, the mutants we see might be a random selection of possible mutants (e.g., all mutants previously observed in any animal have equal possibility of being seen). Or, more likely, there is a hierarchy of likely escape mutations (determined by the mutation probability, fitness etc.), but these probabilities are similar across different animals (so the probability of a given mutation is similar across animals). An alternative scenario is that escape mutant virus (previously observed in plasma in a given animal) rebounded from the reservoir. In this case, the rebounding virus in a given animal should resemble both the mutations and barcodes observed earlier in infection. To test this, we fit a Poisson model of different sources of viral variability early following the 2$^{nd}$ ATI; 1) equal probability of all mutants, 2) observed hierarchy of mutants across all animals, or 3) hierarchy of mutations seen at the time of the second ART in the same animal, to see which best predicted the escape mutants present in plasma. We then explored whether a combination of two mechanisms improved the fit (hypothesis 4). As derived in S3 Text, the probability of variant $k$ being detected following rebound in animal $j$ ($z_{k,j}$) can be modeled as

$$z_{k,j}(\overrightarrow{x}_{k,j}; \overrightarrow{\theta}_j) = 1 - e^{-r_{k,j}(\overrightarrow{x}_{k,j}; \overrightarrow{\theta}_j)} \tag{2}$$

where $r_{k,j}$ is a function of both variant specific and animal specific variables, $\overrightarrow{x}_{k,j}$, and animal specific parameters, $\overrightarrow{\theta}_j$, and takes on a different form depending on the hypothesis being considered (see S3 Text for the 4 forms of $r_{k,j}(\overrightarrow{x}_{k,j}; \overrightarrow{\theta}_j)$).

For each hypothesis (and therefore each form of $r_{k,j}$), Eq (2) was optimized to the data from each animal using a maximum likelihood approach with the log-likelihood for animal $j$ defined by:

$$\ln \mathcal{L}_j(\overrightarrow{\theta}_j) = \sum_k d_{k,j} \ln(z_{k,j}(\overrightarrow{x}_{k,j}; \overrightarrow{\theta}_j)) + (1 - d_{k,j}) \ln(1 - z_{k,j}(\overrightarrow{x}_{k,j}; \overrightarrow{\theta}_j)) \tag{3}$$

where $d_{k,j}$ is an indicator variable for if variant $k$ was detected (1) or not (0) in the reactivation pool of animal $j$, and $\overrightarrow{\theta}_j$ and $\overrightarrow{x}_{k,j}$ are as defined above. The Akaike Information Criterion [42] and likelihood ratio tests were used to compare the relative support for hypotheses 1 to 4 listed above and hence, whether it is most likely that the variants detected following the second treatment interruption were the result of reactivation from the reservoir and/or mutation from reactivated wild-type virus.

**Analysis definitions.** Peak viral load is defined as the maximum viral load within 30 days of detected viral rebound.

The mean time weighted $\log_{10}$ viral set point is calculated based on viral loads between 30 days post rebound detection and 60 days post rebound detection, resumption of ART, CD8 depletion, or necropsy, whichever came first. That is, mean time weighted $\log_{10}$ viral set point is

$$\frac{1}{t_f - t_i} \int_{t_i}^{t_f} \log_{10} V(t) dt$$

where $V(t)$, is the viral load at time $t$, $t_i$ is the time of the first viral load quantification 30 days or more after rebound detection, and $t_f$ is the minimum of 1) time of last viral load quantification within 60 days of rebound detection, 2) time of ART resumption, 3) time of CD8 depletion, and 4) necropsy. For a number of animals, $t_i = t_f$, and in these instances, mean time weighted $\log_{10}$ viral set point was set as $\log_{10} V(t_i)$. It is assumed that $\log_{10} V(t)$ changed linearly between viral load measurements.

Fold increase in viral load following CD8 depletion, is the ratio of first viral load quantification after the day of CD8 depletion to that on the day of CD8 depletion.

When required in calculations, below detection viral loads were set to the limit of detection (60 copies/ml).

Barcodes were deemed to have reactivated post-CD8 depletion as follows. 1) In the 6 animals with sequencing done both prior to and after CD8 depletion, barcodes detected post-CD8 depletion but not earlier in the second interruption were categorized as reactivating post-CD8 depletion. 2) In 3 animals sequenced post-CD8 depletion but not prior (2 animals that did not reactivate prior to CD8 depletion and one that had undetectable viral load for 30 days prior to CD8 depletion), all barcodes detected post-CD8 depletion were categorized as reactivating post-CD8 depletion.

**Statistical packages.** Statistical tests were performed in MATLAB (MathWorks, R2021b), GraphPad Prism 9.4.1, or R version 4.0.3 run through RStudio.

## Supporting information

**S1 Text. Limit of detection for Illumina sequencing.**
(DOCX)

**S2 Text. Tat-SL8-specific CD8+ T cell phenotype flow cytometry analysis.**
(DOCX)

**S3 Text. Modeling predictors of variant reactivation at second treatment interruption.**
(DOCX)

**S1 Table. Tat-SL8-specific CD8+ T Cell Phenotype Comparison: Prior to 1st ATI vs. Prior to 2nd ATI (day ~190 vs. 373–374).**
(DOCX)

**S2 Table. Non-wild-type Tat-SL8 variants detected at greater than 20% of viral load at least once.**
(DOCX)

**S1 Fig. Gating strategy used for flow cytometric analyses.**
(DOCX)

**S2 Fig. Rebounding barcodes.**
(DOCX)

## Author Contributions

**Conceptualization:** Steffen S. Docken, Brandon F. Keele, Mirko Paiardini, Michael R. Betts, Katharine J. Bar, Miles P. Davenport.

**Data curation:** Steffen S. Docken, Kevin McCormick.

**Formal analysis:** Steffen S. Docken, Mykola Pinkevych.

**Investigation:** Kevin McCormick, M. Betina Pampena, Sadia Samer, Emily Lindemuth, Elise G. Viox.

**Methodology:** Steffen S. Docken, Mykola Pinkevych, Elise G. Viox, Yuhuang Wu, Timothy E. Schlub, Deborah Cromer, Mirko Paiardini, Michael R. Betts, Katharine J. Bar, Miles P. Davenport.

**Resources:** Mirko Paiardini, Michael R. Betts, Katharine J. Bar, Miles P. Davenport.

**Writing – original draft:** Steffen S. Docken, M. Betina Pampena, Elise G. Viox, Miles P. Davenport.

**Writing – review & editing:** Steffen S. Docken, M. Betina Pampena, Elise G. Viox, Deborah Cromer, Brandon F. Keele, Mirko Paiardini, Michael R. Betts, Katharine J. Bar, Miles P. Davenport.

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
