## [Decision Letter · Decision Letter 0]

14 Aug 2023

Dear Prof. Davenport,

Thank you very much for submitting your manuscript "Preferential selection of viral escape mutants by CD8+ T cell ‘sieving’ of SIV reactivation from latency" for consideration at PLOS Pathogens. As with all papers reviewed by the journal, your manuscript was reviewed by members of the editorial board and by several independent reviewers. In light of the reviews (below this email), we would like to invite the resubmission of a significantly-revised version that takes into account the reviewers' comments.  In particular, it will be important to fully address the concerns raised by Reviewer 1.

We cannot make any decision about publication until we have seen the revised manuscript and your response to the reviewers' comments. Your revised manuscript is also likely to be sent to reviewers for further evaluation.

Sincerely,

Robert F. Siliciano

Academic Editor

PLOS Pathogens

Susan Ross

Section Editor

PLOS Pathogens

Kasturi Haldar

Editor-in-Chief

PLOS Pathogens

orcid.org/0000-0001-5065-158X

Michael Malim

Editor-in-Chief

PLOS Pathogens

orcid.org/0000-0002-7699-2064

Reviewer's Responses to Questions

**Part I - Summary**

Reviewer #1: This paper examines the role of the cellular immune system in the kinetics of viral rebound in SIV-infected macaques. The main claim of the paper is that it provides evidence that the CD8 T cell (CTL) response provides a selection pressure on viruses reactivating and exiting the latent reservoir, and that in the presence of such a response latent virus with CTL escape mutations preferentially are about to reactivate and contribute to rebound. The study involves 14 animals who are infected then treated with combination ART after 14 days, then after ~7 months treatment is stopped and viral rebound is allowed to occur ("analytic treatment interruption" or ATI). ART is restarted after ~8 weeks, then interrupted AGAIN after about 3 months and rebound allowed to occur AGAIN. Finally, some animals are then subjected to CD8 T cell depletion after the second ATI. The authors are able to dissect rebound kinetics and immune escape because of two important innovations of the study design - a) the virus animals are infected with is barcoded, which allows some understanding of the number of unique viral reactivation events that contribute to rebound, and b) the animals are breed that has a specific HLA background that leads to a CTL response that predominantly targets a single known epitope (thus, they know the specific genetic locus responsible for immune escape and so can easily classify viruses as escape or not, at least at that epitope). The main finding was that even though the viral reservoir (measured by all SIV DNA) was predominantly non-escape virus, escape virus was selected during later rebounds. During the first ATI/viral rebound, early rebounding virus had the same mainly-non-escape genotype as the reservoir, but then escape virus increased in frequency before ART was restarted. During the second rebound, fewer unique reactivating lineages were compared to the first, and, escape virus was at a much higher frequency.

Interestingly, the number of measured peripheral CTL cells specific to the epitope of interest did not actually change from before the first ATI (containing predominantly non-escape virus) to before the second ATI (predominantly escape virus), which suggested that it was some aspect of the "quality" of this response, not just "quantity", that exerted the strong selection pressure on reactivating virus.

In addition, when the authors conducted a CD8 depletion during the second ATI, they found an increase in viral linages without escape mutations, again providing support that CTL pressure was preventing rebound of non-escape strains.

At first the study design seemed a bit odd and convoluted to me (why not just wait longer to start ART, so better immune response built up, and then do a single ATI), but it seems originally these animals were used to test an immunotherapy that turned out not to work and then the data was repurposed for this study (which I applaud the authors for doing - it was just a bit confusing until this was revealed later on in the paper).

Overall I think this is an interesting and rigorous study and the paper is really clearly written with nice graphics etc. It is properly placed in the context of previous literature. I did not find the results to be particularly surprising, and while interesting scientifically, it is unclear how they will push the HIV cure field forward, or what relevance they might have outside it. We know that CTL responses are important for (even imperfect) control of HIV/SIV so naturally they will have some impact on viral rebound. I don't think it's surprising that CTLs have some effect immediately after viral reactivation (it's a memory response, afterall!) and immediately exert a selective pressure. Just because these responses are not strong enough fully prevent rebound, or prevent superinfection, does not mean they do not have any effect.

Reviewer #2: In this report by Docken et al, the authors use the SIV infected Mamu-A01 rhesus macaque model to examine the interaction between CD8 T cells and virus during ART and multiple ART interruptions. The study is highly and narrowly focused on CD8s that target the Tat SL8 epitope, which has been studied extensively for years and has a well defined pattern of viral escape as well as available reagents (tetramers) for focused examination of CD8s that target this epitope. They use a series of two ATIs, and a final CD8 depletion and ultimately find that breakthrough virus during the second ATI is almost entirely comprised of escape variants, despite the majority of sequenceable proviruses remaining wild type. They conclude that CD8s against this epitope created a sieve effect wherein the only virus that could outgrow contained escape mutations. This conclusion was backed by a subsequent CD8 depletion, and the clever use of a barcoded virus that allowed tracking of individual clones and linking them to specific sequences within the SL8 epitope. The study is excellently done and clearly presented. Every question or caveat that occurred to me while reading was either addressed by the next experiment, or clearly discussed as a caveat. There are indeed caveats – notably the early ART initiation, the focus on one epitope, and the lack of tissue sampling, but these are all addressed and the story remains intact and important. I have no significant suggestions for improvement.

Reviewer #3: The authors studied the impact of CD8 T cells on the characteristics of rebounding virus following treatment interruptions in a non-human primate model. Using viral barcode lineages, the authors assessed the evolution of immune escape at the Tat-SL8 epitope following two separate ATIs.

While in the first ATI, wt virus dominated the rebound, following the second ATI, viral rebound was dominated by Tat-SL8 escape mutants. Interestingly, the authors speculate that CD8 T cells act as a “sieve” of the latent reservoir and suppress the reactivation of wt virus in the second ATI. Interestingly, CD8 depletion led to greater wt reactivation highlighting the importance of CD8 immune pressure in reservoir activation. Furthermore, the authors speculate that boosted CD8 T cell function during viral replication in the first ATI may lead to enhanced immune control of wt virus in the second ATI. However, the viral replication of wt in the first ATI may result in viral escape from the CD8 immune response in the second ATI.

The analysis is very interesting, and the manuscript is very well written. Methods are described clearly. I have only minor comments.

**Part II – Major Issues: Key Experiments Required for Acceptance**

Reviewer #1: None

Reviewer #2: (No Response)

Reviewer #3: None

**Part III – Minor Issues: Editorial and Data Presentation Modifications**

Reviewer #1: I found that the calculation of "viral reactivation rate" was confusing and unnecessary for the message of the paper. When I tried to understand this calculation I found the authors mainly referred to a previous paper, which is very annoying for a reader. Importantly, they did not even summarize the assumptions that went into the calculation (which definitely should be included in a revision). I tried to briefly look at the referenced papers and it seems the calculation requires assuming that reactivation events are evenly spaced, and that the growth rates of all strains identical . Both these seem unlikely to be true, but the second especially ... the whole point of this paper is showing that growth rate of some non-escape strains is so much lower than that of escape strains that none are detected at all! So naturally this would not be a binary thing but likely continuous and there could be a while distribution of rates. I thought the paper would be stronger if the authors used some measure of number of reactivating virus or other quantity extracted from the barcodes that was more data-driven and less reliant on an unclear and perhaps overly restrictive model.

In a similar vein, I also thought it would be more clear if the authors reminded readers somewhere that their measures of # of reactivating lineages were only those surviving, and that many others could have died out by stochastic extinction before detection.

In addition I thought it was an interesting observation that even though their was more evidence of CTL escape at the second ATI, there was still lower peak and set point viral load than the first ATI, suggesting that overall control of viremia was higher, perhaps due to the combined effect of a fitness cost of the escape mutation(s) and the residual CTL control. This is important because people might (falsely) assume that using future immunotherapies to control rebound could make things worse than no treatment if they select for escape (like people sometimes think about drug resistance). Of course this might be case specific but it clearly doesn't seem like the case here.

I thought the authors could give a little bit more background on this particular macaque model they were using and why they feel they can assume escape vs non-escape strain based on a single epitope, as readers unfamiliar with that literature would be a little lost here.

This is probably a better fit for future work, but I also thought the study would be a good opportunity to do some modeling to provide some quantitative estimate of the cost of escape/strength of CTL response, rate of CTL expansion, etc.

Finally, in this day and age I don't think it's acceptable to submit a paper for review with a data/code availability statement saying it will be shared upon publication. This should be part of the review process. When I was looking back at the authors' previous paper in this same journal to see the methods about the rebound virus (Fennessy 2017) I noticed there they mistakenly claim their data is available in the SI but it is not, so these resources need to be confirmed by reviewers/editors pre-publication.

Reviewer #2: The abstract is unnecessarily short for this journal, and leaves out important details of the study. This is a minor criticism at best.

Reviewer #3: Figure 1: Sub-figures C, E, and G are of a poor quality. Please enhance the quality of the light and dark green dots and lines.

Figure 1B caption 1: There is a typo in line 125.

Line 146: Since FTY720 had no effect on the rebound characteristics, was there any difference between FTY720 and control group? Has it been previously shown that the FTY720 had a limited or no effect? Maybe you could add a reference.

Figure 2: It seems that the mutant increases in two phases: initially fast and later slow. What do you think drives the two-phasic increase in mutant virus? Maybe you can write one or two sentences in the manuscript explaining the phenomenon.

Line 513/514: You could refer to Figure 1B.

Line 562/563: closing bracket is missing.

PLOS authors have the option to publish the peer review history of their article (what does this mean?). If published, this will include your full peer review and any attached files.

Reviewer #1: No

Reviewer #2: No

Reviewer #3: No
---

## [Editor Report · Decision Letter 1]

15 Oct 2023

Dear Prof. Davenport,

We are pleased to inform you that your manuscript 'Preferential selection of viral escape mutants by CD8+ T cell ‘sieving’ of SIV reactivation from latency' has been provisionally accepted for publication in PLOS Pathogens.

Best regards,

Robert F. Siliciano

Academic Editor

PLOS Pathogens

Susan Ross

Section Editor

PLOS Pathogens

Kasturi Haldar

Editor-in-Chief

PLOS Pathogens

orcid.org/0000-0001-5065-158X

Michael Malim

Editor-in-Chief

PLOS Pathogens

orcid.org/0000-0002-7699-2064
---

## [Editor Report · Acceptance letter]

27 Oct 2023

Dear Prof. Davenport,

We are delighted to inform you that your manuscript, "Preferential selection of viral escape mutants by CD8+ T cell ‘sieving’ of SIV reactivation from latency," has been formally accepted for publication in PLOS Pathogens.

Best regards,

Kasturi Haldar

Editor-in-Chief

PLOS Pathogens

orcid.org/0000-0001-5065-158X

Michael Malim

Editor-in-Chief

PLOS Pathogens

orcid.org/0000-0002-7699-2064